# De Novo Assembly, Characterization and Comparative Transcriptome Analysis of the Gonads of Jade Perch (*Scortum barcoo*)

**DOI:** 10.3390/ani13142254

**Published:** 2023-07-10

**Authors:** Shiyan Liu, Yingying Lian, Yikun Song, Qinghua Chen, Jianrong Huang

**Affiliations:** 1State Key Laboratory of Biocontrol, Institute of Aquatic Economic Animals and Guangdong Provincial Key Laboratory for Aquatic Economic Animals, School of Life Sciences, Sun Yat-Sen University, Guangzhou 510275, China; 2South China Institute of Environmental Science, MEE, Guangzhou 510610, China

**Keywords:** *Scortum barcoo*, jade perch, transcriptome, sex-related genes, gonad differentiation and development

## Abstract

**Simple Summary:**

Jade perch (*Scortum barcoo*) has been affirmed as an economically important species for aquaculture. However, research on its sex and reproductive regulation is still lacking. Herein, the first gonad transcriptomes of juvenile jade perch were identified through Illumina Novaseq technology. According to the comparative transcriptome data of ovaries and testis, a large number of differential expression genes (DEGs) mainly involved in the steroidogenesis pathway, gonad development and differentiation, gametogenesis and gamete maturation were identified. As predicted, the expression patterns of these DEGs in jade perch were similar to well-known sex-related genes in other species. These results will provide valuable information for further studies on sex determination and gonadal development in teleost fish.

**Abstract:**

Due to the high meat yield and rich nutritional content, jade perch (*Scortum barcoo*) has become an important commercial aquaculture species in China. Jade perch has a slow growth rate, taking 3–4 years to reach sexual maturity, and has almost no difference in body size between males and females. However, the study of its gonad development and reproduction regulation is still blank, which limited the yield increase. Herein, the gonad transcriptomes of juvenile males and females of *S. barcoo* were identified for the first time. A total of 107,060 unigenes were successfully annotated. By comparing male and female gonad transcriptomes, a total of 23,849 differentially expressed genes (DEGs) were identified, of which 9517 were downregulated, and 14,332 were upregulated in the testis. In addition, a large number of DEGs involved in sex differentiation, gonadal development and differentiation and gametogenesis were identified, and the differential expression patterns of some genes were further verified using real-time fluorescence quantitative PCR. The results of this study will provide a valuable resource for further studies on sex determination and gonadal development of *S. barcoo*.

## 1. Introduction

Jade perch (*Scortum barcoo*), also known as barcoo grunter, belongs to the genus *Scortum*, the family Theraponidae and the order Perciformis [1]. It is indigenous to Australia and normally found in the Lake Eyre Basin and Barcoo River of Australia [2]. Jade perch has a high meat yield and rich nutritional value. Currently, the main sources of omega-3 are deep-sea fish and algae, which are difficult to obtain. Thus, jade perch is believed to be an alternative source of omega-3 due to its high level of omega-3 [3]. Moreover, jade perch grow rapidly and can reach commercial specification in 6–10 month under artificial aquaculture conditions [4]. Jade perch adapt well to high-density aquaculture and are suitable for cage culture and factory recirculating aquaculture [5]. Since being introduced into China in 2001, it has become an extremely important and valuable fish in China [6]. Under natural conditions, jade perch can reach sexual maturity at 3–4 years old. In the non-breeding season, the individual differences between mature male and female fish are small. During the breeding season, the belly of the jade perch accumulates a lot of fat, inducing abdominal enlargement, so it is also difficult to distinguish its sex from the external morphology. The lack of sexual dimorphism in body size and the inability to discriminate between males and females based on a single trait of jade perch greatly limit the selection for reproductive production, sex control and genetic improvement [7,8], which limits the yield increase of jade perch. Thus, a better understanding of the sex-related genes of this species should be developed.

Stephan first proposed the universal phenomenon of sex transition is a common phenomenon in fishes [9]. Later, researchers found sex transition in *Monopterus javanensis* and Sparidae fishes [10,11], which started the research on sex determination and sex transition in fishes. The sex of fish is regulated by both genetic and environmental factors [12]. Unlike environmental factors, genetic factors determine the initial sex of fish [13]. After sex determination, the development process of sex-related traits also involves more complex gene interactions in which sex-differentiation genes play a key role [14]. At present, many sex-determining genes and sex-determination pathways have been identified in fish, including *dmy*, *sdY*, *amhr2*, *amhy*, *gsdf*, *dmrt1*, the *Wnt* signaling pathway, the transforming growth factor-β (*TGF*-*β*) gene family and the estrogen signaling pathway [15,16]. In the process of aquaculture, it is of great significance to study the mechanism of sex control in fish. First, it is beneficial to achieve the artificial control of fish sex at the biological molecular level in production, which is crucial for reproductive management and breeding practices. Secondly, it is of great significance to clarify the theories of sex determination and differentiation in fish. Up to now, previous studies mainly focused on breeding, morphology, rearing conditions, feeding efficiencies and immune responses of jade perch [4,5,7,17,18,19,20], whereas less attention was paid to sex determination and differentiation. Therefore, it is necessary to explore functional genes related to sex determination and differentiation in jade perch in order to better understand the mechanisms of sex regulation and effectively guide jade perch breeding for sex control.

Next-generation sequencing (NGS) based on transcriptome sequencing technology has been widely used to provide gene expression profiles and regulatory mechanisms in specific tissues or organs [21,22]. Multiple aquaculture species such as Nile tilapia (*Oreochromis niloticus*) [23], spotted scat (*Scatophagus argus*) [24], spotted knifejaw (*Oplegnathus punctatus*) [16], Chinese tongue sole (*Cynoglossus semilaevis*) [25], fugu (*Takifugu tubripes*) [26], silver sillago (*Sillago sihama*) [27], channel catfish (*Ictalurus punctatus*) [28] and spot-fin porcupine fish (*Diodon hystrix*) [29] have been identified as having a large number of sex-specific genes and male-female regulatory pathways using gonad transcriptome sequencing. The available genetic information on jade perch is rather limited, with only a few studies on the genome and mitochondrial genome of jade perch [6,8,30]. To date, few sex-related genes have been studied in jade perch, and basic information on gonadal transcriptome expression is still lacking.

In this study, the first comparative transcriptome analysis of jade perch gonads was generated using Illumina-based transcriptome sequencing. The transcriptome comparative analysis results revealed a great number of sex-related genes, which were further identified and discussed following quantitative real-time PCR (qRT-PCR). Our study aims to fill the blanks in jade perch gonadal transcriptome data, identify candidate sex-related genes and provide a theoretical basis for the subsequent exploration of jade perch sex-determining genes.

## 2. Materials and Methods

### 2.1. Sample Collection

In our study, 6 seven-month-old jade perch individuals (3 females and 3 males), the gender of which was identified by the morphological observation of the gonads, were obtained from Shaoguan, Guangdong province. The fishes were anesthetized with 300 mg/L MS222 (Sigma, Saint Louis, MO, USA). Gonads from each individual were excised within 60 s of euthanasia. The gonads to be used for RNA extraction were stored in liquid nitrogen, and those intended for gonadal histology analysis were fixed in Bouin’s solution for 24 h at room temperature and then transferred to 75% ethanol. All animal experiments were conducted in accordance with the guidelines and approval of the respective Animal Research and Ethics Committees of Sun Yat-Sen University.

### 2.2. RNA Extraction and Library Construction

The total RNA was extracted using an RNA-isolating Total RNA Extraction Reagent (Vazyme, Nanjing, China), following the manufacturer’s instructions. The quantity and quality of the RNA were examined using Nanodrop 2000 (Thermo Scientific, Wilmington, DE, USA) and 1.0% agarose gel electrophoresis, respectively. Then, the integrity and quantity of the RNA were checked with a precise concentration using an Agilent 4200 Bioanalyzer (Agilent Technologies, Santa Clara, CA, USA). RNA with qualified quality was entered into the library construction process. The jade perch transcriptome libraries were constructed using the NEBNext^®^ Ultra™ RNA Library Prep Kit for Illumina^®^ (NEB, Ipswich, MA, USA) kit. The mRNA of jade perch was enriched using magnetic beads with Oligo (dT). Under a high-temperature environment and the presence of metal ions, the RNA was fragmented, and the first cDNA strand was synthesized by random hexamers. Then the second cDNA strand was synthesized by adding an enzyme, a buffer solution and dNTPs (dATP, dTTP, dGTP and dCTP). The synthesized double-stranded cDNA was purified using magnetic beads, followed by end repair and the addition of A. The ligation of sequencing adapters and fragment size sorting was performed using magnetic sorting beads. The sorted fragments were enriched by PCR, and the final PCR products were purified to obtain the final library.

### 2.3. Library Sequencing, De Novo Assembly and Annotation

The qualified libraries were sequenced using the Illumina Novaseq6000 (Illumina, Inc., San Diego, CA, USA) high-throughput sequencing platform with a sequencing strategy of PE150 (Pair-End 150), and the amount of sequencing data for each gonadal sample was not less than 6 Gb.

The results of Illumina high-throughput sequencing were converted to raw reads after base calling using CASAVA (v1.8.2) software. In order to filter the splice sequences and low-quality sequences in the raw data and ensure the quality of analysis data, Fastp v0.20.1 software [31] was used to filter the raw reads, and high-quality clean reads were obtained. The parameters for filtering data in Fastp v0.20.1 software were set to “-l 150 -q 20”, and the rest were default parameters. All subsequent analyses were based on clean reads. Then, the de novo assembly was carried out by Trinity (v2.4.0) software [32]. The integrity of sequence assembly was assessed by BUSCO v5.2.2 [33].

Three forward and 3 reverse reading frames were used to predict the coding region of the unigene, and then 6 possible coding protein sequences were predicted. The obtained coding protein sequences were compared with non-redundant protein sequences (Nr) (https://www.ncbi.nlm.nih.gov/, accessed on 20 July 2022) and the UniProt protein database (https://www.uniprot.org/, accessed on 25 July 2022). Finally, the encoding method with the best matching (the maximum score) was selected as the encoding method for this gene.

Based on homology searches, unigenes were annotated against major public databases, including the Clusters of Eukaryotic Orthologous Group database (KOG; http://www.ncbi.nlm.nih.gov/KOG, accessed on 28 June 2022), the Nr database, the UniProt protein database and the Kyoto Encyclopedia of Genes and Genomes (KEGG; http://www.genome.jp/kegg, accessed on 18 July 2022) using BLASTP or Diamond. Blast2GO was used to further annotate the Nr results [34]. The Gene Ontology (GO; http://geneontology.org, accessed on 20 July 2022) analysis was performed according to the function of the protein sequence. The GO enrichment bar chart could visually reflect the information of genes in 3 aspects: biological process, cellular component and molecular function.

### 2.4. Identification of DEGs and Enrichment Analysis

Using HISAT2 v2.1.0 [35], the clean reads of each gonadal sample are first aligned to the non-redundant assembled transcripts. According to the alignment results, StringTie v1.3.3b [36] was used to further calculate the transcripts per million (TPM), fragments per kilobase of exon model per million mapped fragments (FPKM), coverage and other gene-expression-related values of each transcript. Python prepDE.py was used to convert the output results of StringTie into the “edgeR” package (V3.6) recognition format [37,38], and edgeR was used for gene differential expression analysis. “*p* value < 0.05, |log2FC|>2” was used for screening condition to screen significantly DEGs. The number of DEGs was counted by statistical significance, and then against the background of DEGs, the clusterProfiler program of the R package [39] was applied based on Fisher’s exact test and Benjamini correction. The pathways with *p*-values < 0.05 were significantly enriched in DEGs compared with the whole genome.

### 2.5. Validation of DEGs Using Quantitative Real-Time PCR (qRT-PCR)

To validate the reliability of DEGs from RNA-seq, a total of 18 putatively sex-related DEGs were selected for quantitative real-time PCR (qRT-PCR) analysis. The primers of the DEGs and the reference gene *β-actin* were designed using Primer 5.0 software (Table 1). The cDNA templates were synthesized using the HiScript III RT SuperMix for qPCR (+gDNA wiper; Vazyme, Nanjing, China). The qRT-PCR was conducted using the SYBR Green qPCR Mix (GDSBIO, Guangzhou, China) and performed on a Roche LightCycler 480 real-time PCR system (Roche, Basel, Switzerland) as per the following process: pre-denaturation at 95 °C for 3 min followed by 45 cycles at 95 °C for 10 s, 58 °C for 20 s and 72 °C for 15 s and a final extension performed at 72 °C for 5 min, ending with a dissociation curve process. Each gene was tested for 3 independent biological replicates and 3 technical replicates, and the expression of each gene was normalized using the *β-actin* by 2^−ΔΔCT^ method [40].

### 2.6. Gonadal Histology

Firstly, the gonads were successively dehydrated in 85%, 95% and 100% ethanol. After dehydration, the gonads were placed in xylene to clear and embedded in paraffin for 3 h. The embedded tissue blocks were serially sectioned at a thickness of 5–7 μm. After drying at 42 °C overnight, the slices were stained with hematoxylin and eosin. The clear slides were quickly sealed with neutral resin and air-dried overnight at room temperature. Stained gonads sections were observed and photographed using a light microscope (Nikon Eclipse Ni-U, Shinagawa-ku, Japan).

## 3. Results

### 3.1. Overview of Illumina Sequencing and Assembly Results

A total of six cDNA libraries, including three testes and three ovaries, were constructed by RNA-seq. After quality control and filtering, a total of 41.18 Gb of clean reads were generated using the Illumina HiSeq platform, with an average of 6.86 Gb and a range of 5.78–7.92 Gb per sample. The mean value of Q20 was 97.71%, and that of Q30 was 93.46% (Table 2). After being assembled de novo, a sum of 107,060 unigenes was assembled, and the total length of these unigenes was 107,070,530 bp. The sequence length distribution of the unigenes ranged from 185 to 18,090 bp, with an average length and N50 of 1000 bp and 2336 bp, respectively (Table 3). Based on the predicted length statistics of unigenes, a total of 81,664 (76.28%) unigenes were 1–1000 bp in length, and 3202 (2.99%) of them were more than 5000 bp in length (Figure 1).

### 3.2. Unigenes Annotation

To further study the specific molecular processes in jade perch, all of the predictable unigenes were annotated in different databases by significant similarity to protein sequences. Then, a total of 28,514 unigenes were annotated successfully. Among all the annotated unigenes, 28,165 (98.78%) unigenes were matched with the Uniport database, which had the highest percentage of annotated unigenes. However, this was only 15,592 (54.68%) unigenes compared to the KOG database, which may be related to the genomic information available limitation for jade perch (Table 3). Species distribution statistics were performed according to the most similar genes compared to the Nr database. The species with the largest number of genes indicated that the species contained the largest number of genes in the Nr database and was relatively similar to that of jade perch. The results showed that *Siniperca chuatsi* (4985, 17.87%), *Chelmon rostratus* (1836, 6.58%), *Morone saxatilis* (1454, 5.21%), *Larimichthys crocea* (1372, 4.92%) and *Liparis tanakae* (1024, 3.67%; Figure 2).

In addition, the GO, KOG and KEGG databases were used for functional prediction and classification of all unigenes. In the GO database, a total of 71,984 (67.24%) unigenes were annotated and classified into three functional categories. Among these three functional groups, the terms “cellular process” (36.15%), “cellular anatomical entity” (88.10%) and “binding” (45.87%) were dominant in the biological process, cellular component and molecular function aspects, respectively (Figure 3A). Based on the KEGG database, a sum of 18,855 unigenes was clustered into five different functional categories. The top three distributions were “signal transduction” (2755 unigenes), “global and overview maps” (1742 unigenes) and “endocrine system” (1398 unigenes; Figure 3B). After KOG annotation, a total of 13,725 unigenes were grouped into 25 classifications, and the largest distribution was “general function prediction only” (2835 unigenes), followed by “signal transduction mechanisms” (2591 unigenes). 

### 3.3. Differential Genes Expression Analysis

The TPM values were used to normalize the levels of gene expression. A total of 23,849 DEGs were identified between the testes and ovaries, with 14,332 (60.09%) DEGs significantly high expressed in testes and 9517 (39.91%) high expressed in ovaries (Table 4). The DEGs information is shown in the volcano plot (Figure 4). The enrichment analysis of these DEGs was further performed using GO and KEGG pathway analyses. The results of the KEGG enrichment analysis revealed that the neuroactive ligand-receptor interaction was the most representative, followed by the calcium signaling pathway and the cytokine-cytokine receptor interaction (Figure 5).

Moreover, numerous sex-related genes were identified based on the GO and KEGG annotation (Table 5). Genes such as growth differentiation factor 9 (*gdf9*), bone morphogenetic protein 15 (*bmp15*), factor in the germline alpha (*figla*) and protein nanos 3 (*nanos3*) were highly expressed in the ovaries. Meanwhile, anti-mullerian hormone (*amh*), insulin-like growth factor 1 (*igf1*), follicle-stimulating hormone receptor (*fshr*), protein nanos 2 (*nanos2*) and so on were highly expressed in the testes (Table 5).

### 3.4. Validation of Transcriptomic Results by qRT-PCR

A sum of 10 testes-biased and eight ovaries-biased genes was selected to validate the transcriptomic expression profiles by qRT-PCR analysis. As expected, the expression patterns of qRT-PCR were consistent with the RNA-seq results (Figure 6). DEGs such as *hsd11b1*, *gsdf*, *sf1*, *dmrtb1*, *dmrt3*, *nanos2*, *amh*, *sox9*, *dscam* and *cyp11a1* were ovaries-upregulated (Figure 6A), whereas unigenes such as *figla*, *hsd17b1*, *dmrt2b*, *bmp15*, *nanos3*, *foxl2*, *foxh1* and *gmnn* were testes-upregulated (Figure 6B). To validate the reliability and accuracy of DEG transcriptomic expression levels, the consistent tendencies of expression levels between the RNA-Seq and qRT-PCR data were evaluated using a correlation analysis (Figure 6C).

### 3.5. Gonads Histology Analysis

The testes and ovaries of 7 month old jade perch were used for the histological analysis. The testes of jade perch were full of primary spermatogonia, which did not show proliferation or differentiation, and no spermatocyte was found (Figure 7A). Meanwhile, primary growth oocyte appeared in the ovary (Figure 7B).

## 4. Discussion

It is believed that sex determination is the most paramount for successful reproduction in animals. During the process of sex determination, a set of sexual functional genes is involved in many complex biological processes to support the gonads’ differentiation into ovaries or testes. To figure out the complicated regulatory mechanism of sex determination, transcriptome sequencing has been demonstrated in more and more species and is considered a highly effective method. Jade perch, which has a high meat yield and rich nutritional value, is a significant commercial aquaculture species. Up to now, the gene expression profile and the molecular mechanisms during sex determination and gonad differentiation have not been revealed in jade perch. In our study, transcriptome analysis and comparative analysis have been conducted between 7 month old male and female juveniles and the expression patterns of DEGs in the differentiating gonads were established for jade perch.

### 4.1. DEGs Involved in Steroidogenesis Pathway

In fish, sex steroid hormones, especially 17β-estradiol (E_2_) and 11-ketotestosterone (11-KT), are involved in sex determination, sex differentiation and sex maintenance [41,42]. Due to the different expression profiles of genes that encode steroid-metabolizing enzymes, the serum levels of E_2_ and 11-KT have sexual dimorphism between females and males [24,43]. At present, *cyp19a1a*, *cyp11b2*, *cyp11a2*, *hsd17b* and *hsd11b1* are considered essential steroidogenesis genes [44].

As a key enzyme in the steroid hormone synthesis pathway, StAR is responsible for transporting cholesterol, the initial substrate for steroid synthesis, from the cytoplasm to the mitochondria [45]. Changes in the expression of *StAR* are able to modulate various signals of steroidogenesis, enabling a rapid response by steroidogenic tissues. In fish, *StAR* is thought to play a role in testicular development, spermatogenesis and spermatogenesis by regulating androgen production. The expression of the *cyp11b2* gene, serum testosterone and 11-KT levels in *StAR2* knockout male tilapia were significantly lower than those in control fish [46]. Mutant *StAR1* in zebrafish significantly increases testosterone and estrogen levels and inhibits oocyte maturation [47]. In jade perch, the expression of the *StAR1* and *StAR2* genes was higher in the testes, which agreed with the expression profile of spotted scat and bastard halibut (*Paralichthys olivaceus*) [24,48]. 

In male fish, androstenedione is catalyzed by 17β-HSD (*hsd17b3*), P450c11 (*cyp11b2*) and 11β-HSD (*hsd11b2*) to form 11-KT, while in female fish, androstenedione is converted to E_2_ by P450c19 (*cyp19a1a/b*) [41,49,50]. As a crucial gene encoding key rate-limiting enzymes that catalyze the synthesis of fish-specific 11-KT, *cyp11b2* plays an important role in the development of the testes. *Cyp11b2* was found to be highly expressed during spermatogenesis in catfish (*Clarias gariepinus*), rainbow trout (*Oncorhynchus mykiss*), Japanese eel (*Anguilla japonica*), tilapia, Atlantic salmon (*Salmo salar*) and medaka (*Oryzias latipes*), but hardly detectable at all stages of ovarian development, suggesting that *cyp11b2* plays an important role in spermatogenesis [51,52,53,54,55,56,57]. Moreover, in the development of fish ovaries, the *cyp19a1a* gene, which can directly catalyze the switching of androstenedione to E_2_, plays an essential role in the process of primitive gonads differentiating into ovaries [58,59]. After the expression of the mutant *cyp19a1a* gene in the female medaka, the gonads still develop into ovaries in the early stage, but a partial female-to-male sex reversal occurs in the later stage [60]. In zebrafish and Nile tilapia, mutant *cyp19a1a* results in female-to-male sex reversal [61,62,63,64]. Similarly, the expression of *cyp11b2* in males was higher than that in females in jade perch, whereas the *cyp19a1a* gene was highly expressed in the ovaries. Therefore, the results of these genes indicated that these DEGs were crucial in the steroid hormone synthesis pathway and gonad development in fish.

### 4.2. DEGs Involved in Gonad Differentiation and Development

Sex determination is influenced by genetics, the environment or both. The development of testes and ovaries is crucial for gametogenesis and reproduction. Therefore, understanding the molecular mechanisms of sex determination is an important direction in reproductive biology. Sex determination and gonadal differentiation in fish are diverse. In recent years, nearly 20 different genes have been confirmed to be involved in teleost sex determination and gonadal differentiation, including the Dmrt gene family, the Sox gene family and the transforming growth factor-β (TGF-β) gene family [15,16]. 

The *Dmrt1* gene is specifically expressed in the male gonads of semi-smooth tongue sole at the critical stage of sex determination, and its expression level is significantly up-regulated in phenotypic males after female sex reversal [65]. *Dmrt1* deficient Thai betta fish (*Betta splendens*) undergo sex reversal from male to female [66]. *Steroidogenic factor*-*1* (*sf1*) is a key gene in vertebrate sex determination. It is expressed in the adrenal gland, gonad, spleen, ventromedial hypothalamus and pituitary gonadotrophins and regulates a variety of steroidogenic enzymes, including aromatase [67]. Protandrous black porgy fish (*Acanthopagrus schlegeli*) has a high expression of *sf1* in the testis and a low expression in the ovary before sexual maturity, and it has an antagonistic effect on the development of oocytes [68]. Transcription factor binding sites for *dmrt1* and *sf1* were revealed using a computer-simulated promoter analysis 5′ upstream of the spotted scat *gsdf* gene. Results showed that *dmrt1* activated *gsdf* expression in spotted scat in a dose-dependent manner in the presence of *sf1* [69]. In jade perch, the expression of *dmrt1*, *dmrt3* and *sf1* were higher in testes, which suggests a key role in the testes’ development.

In recent years, members of the TGF-β signaling pathway have also been confirmed by more and more researchers to be sex-determining genes in vertebrates, especially in fish [70]. Mullerian ducts are absent in fish, but the anti-Mullerian hormone (*amh*) gene is widely present in fish and participates in fish germ cell proliferation. In zebrafish, mutations in the *amh* gene have been found to increase the proportion of females [71]. In addition, the *amh* copy gene *amhy* has been confirmed to be the sex-determining gene in Patagonian pejerrey (*Odontesthes hatcheri*), Nile tilapia and Northern pike (*Esox lucius*) by gene editing knockout and transgenic methods [72,73,74]. Myosho et al. found a specific copy of *gsdf* on the Y chromosome in *Oryzias luzonensis*, and the up-regulation of this gene in the reproductive primordium of the early embryo would lead to the differentiation of gonads into males [75]. Through genome sequencing analysis, it was found in sablefish (*Anoplopoma fimbria*) that there were specific sequences of different lengths in the upstream region of the *gsdf* gene on the X and Y chromosomes [76]. Similarly, A specific copy of the *gsdf* gene was found on the Y chromosome of Atlantic halibut (*Hippoglossus hippoglossus*) [77], suggesting that *gsdf* and its adjacent upstream regulatory elements are responsible for sex determination. Growth and differentiation factor 9 (*gdf9*) and bone morphogenetic protein 15 (*bmp15*) are involved in ovarian development in bony fish by regulating the expression of gonadotropin and its receptors. In European sea bass (*Dicentrarchus labrax*) and Japanese flounder, changes in the expression levels of *gdf9* and *bmp15* affect the expression of follicle-stimulating hormone receptors (*fshr*) and luteinuclein receptors (*lhr*) [78,79]. Overexpression of *bmp15* in cultured zebrafish eggs could inhibit gonadotropin-induced ovarian maturation [80]. The expression pattern of these genes in jade perch was in accord with the roles of sex determination, i.e., the mRNA levels of *gsdf* and *amh* were higher in males, while the *bmp15* and *gdf9* genes had higher expression levels in females. 

The *nanos2* gene is specifically expressed in male germ cells [81]. The *Nanos2* gene is specifically expressed in tilapia testes but not expressed in eggs and ovaries [82]. The *nanos3* gene regulates the migration of primordial germ cells and plays an important role in the production and maintenance of oocytes and the development and differentiation of germ cells [83]. Previous studies have confirmed that the *nanos3* gene is highly expressed in the ovary of zebrafish (*Danio rerio*) and large yellow croaker (*Larimichthys crocea*) and involved in the development and maintenance of germline stem cells (GSCs), whereas it is expressed to a lesser extent in the testes [84,85]. In jade perch, *nanos2* expression was higher in males, and *nanos3* expression was higher in females, which suggests its important role in germ cell development and differentiation.

### 4.3. DEGs Involved in Gametogenesis and Gamete Maturation

In the praxis of fish culture, the ultimate goal of breeding farmed species is to promote the development of testes and ovaries and then to obtain healthy mature gametes for the generation of offspring. In recent years, genes including zona pellucida nuclear receptor subfamily 0 group B member 1 (*nr0b1*), forkhead box (*Fox*) and sperm-binding proteins (*zps*) have been identified as associated genes of oogenesis, oocyte maturation, spermatogenesis and sperm motility [21,29].

Dax1 (DSS-AHC critical region on the X chromosome 1, gene 1), which belongs to the nuclear receptor family, is a special orphan protein encoded by the *nr0b1* gene [86,87]. *Dax1* has been identified as an important factor in the regulation of embryonic cell pluripotency [88]. The expression pattern of *dax1* varies among different fish species. It is found that *dax1* begins expression in the bipotential gonads of zebrafish 13 days after hatching and has significant expression in the ovary after the morphological differentiation of the gonads. Moreover, in *dax1*^*−/−*^ zebrafish, oocyte proliferation is reduced, resulting in arrested ovarian development and sex reversal to testes [89]. In medaka, *dax1* is highly expressed in the ovary of adult fish but is hardly expressed in the testes [90]. On the contrary, in tilapia, the *dax1* gene is continuously expressed from the critical period of sex determination and differentiation (5 days post-hatching) into adult fish, and there is no significant difference in the expression level between female and male gonads [91]. Shi et al. proposed that *dax1* is involved in ovarian development in spotted scat by regulating Nr5A1-mediated *cyp19a1a* expression [92]. According to the transcriptome data in this study, the *nr0b1* gene was more highly expressed in jade perch testes, which showed a different pattern.

The Fox family is a class of transcriptional regulatory nuclear proteins, and members of the Fox family are involved in the regulation of multiple biological processes. Researchers indicated that *fox* genes might play important roles in sex determination and gonadal development in teleosts [93]. In zebrafish, *foxh1* expression is higher in females than in males, and zebrafish eggs with *foxh1* gene deletion develop abnormally [94,95]. However, in tilapia, *foxh1* is expressed in the cytoplasm of oocytes in the ovary [93], and female fishes with *foxh1* knockout are arrested in follicular development, which eventually leads to female infertility [96]. In teleost fish, zona pellucid (ZP) protein provides mechanical protection to the oocyte during fertilization, mainly structurally [97,98]. *Zp3*, a member of the *zp* superfamily, is a major female-specific gene that forms the extracellular matrix around the oocyte and mediates sperm binding. The *Zp3* gene is highly expressed in the ovaries of Spot-fin porcupine fish and *Spinibarbus hollandi*, and the researchers hypothesized that *zp3* genes may be involved in ovarian folliculogenesis [21,29]. Herein, the *foxh1* and *zp3* genes had higher expression levels in female jade perch, suggesting the important roles of these genes in oogenesis. 

## 5. Conclusions

This work is the first research on the gonad transcriptome of juvenile jade perch. Herein, we assembled 107,060 unigenes based on gonad transcriptome data. A high number of DEGs, which are mainly involved in the steroidogenesis pathway, gonad development and differentiation, gametogenesis and gamete maturation, were identified by comparative transcriptome analysis. For the proven DEGs associated with gonadal development and reproduction, we found similar expression profiles in jade perch, suggesting that they play a conserved role in gonadal development and gametogenesis. In addition, we also identified and analyzed the expression profiles of other genes that may be involved in gonad development and gametogenesis. The results of this study will provide valuable information for further studies on sex determination and gonadal development in teleost fish.

## Figures and Tables

**Figure 1 animals-13-02254-f001:**
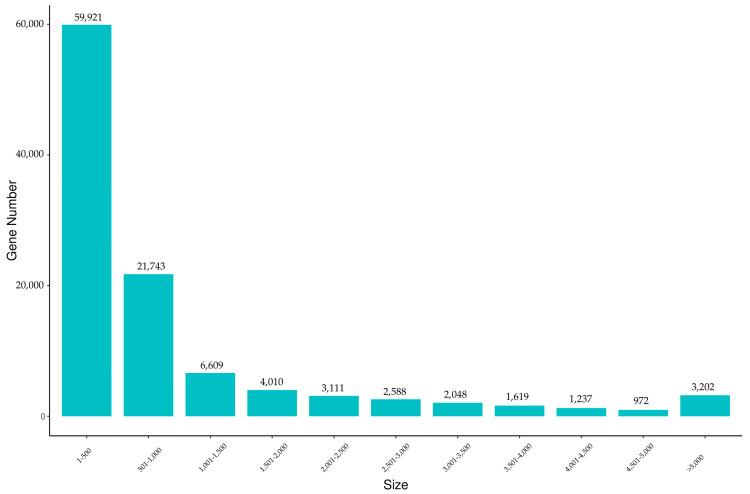
Length distribution of assembled unigenes of the jade perch gonadal transcriptome. *X*-axis: size of unigene; *Y*-axis: number of unigenes.

**Figure 2 animals-13-02254-f002:**
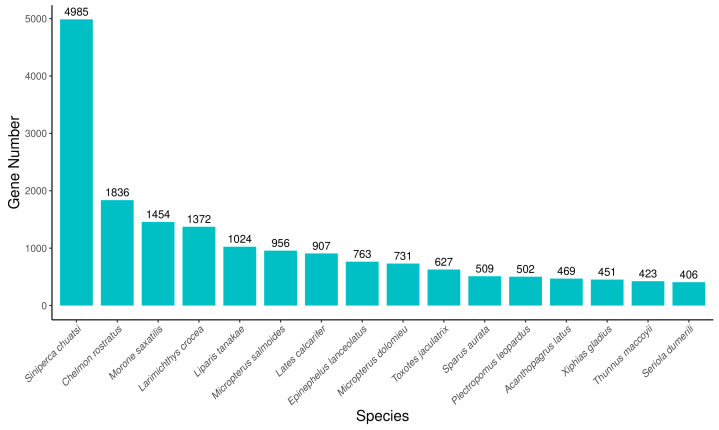
The species distribution of the results of Nr annotation. *X*-axis: the top species which match the annotated sequences distribution. *Y*-axis: the number of annotated sequences matching each species.

**Figure 3 animals-13-02254-f003:**
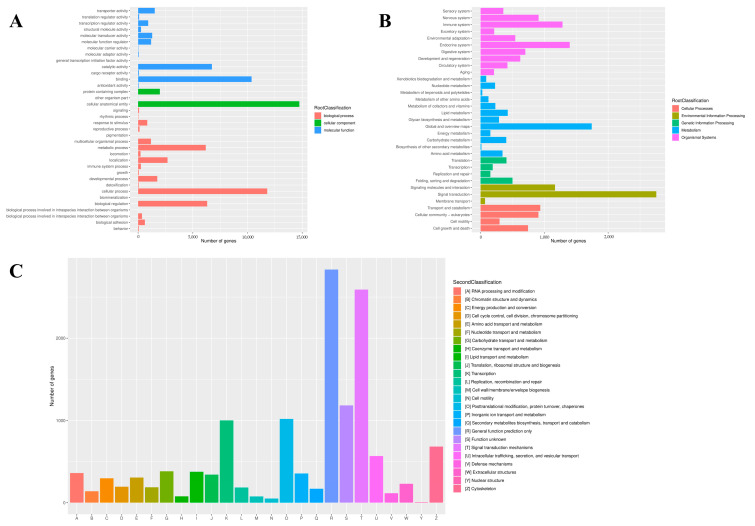
Function annotation of unigenes based on GO (**A**), KEGG (**B**) and KOG (**C**) databases.

**Figure 4 animals-13-02254-f004:**
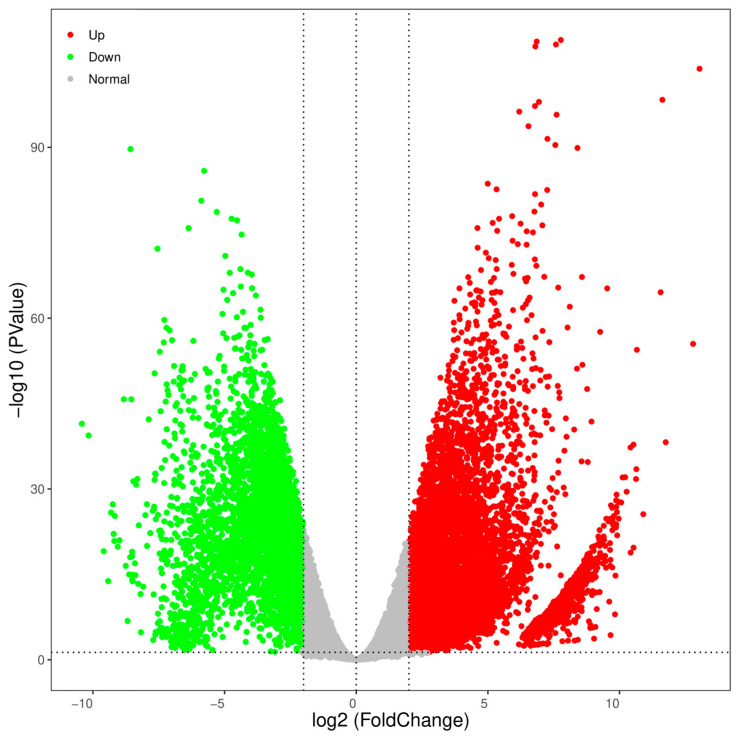
Volcano plot of DEGs in testes versus ovaries. Upregulated and downregulated genes were respectively represented by red and green points in males and vice versa.

**Figure 5 animals-13-02254-f005:**
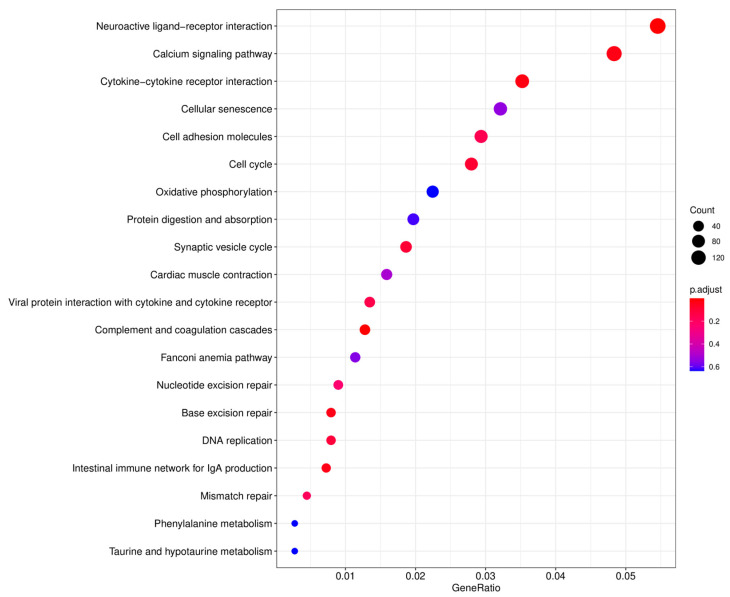
KEGG enrichment analysis of the top 20 pathways. *X*-axis: the ratio of the number of differential genes annotated in the KEGG pathways to the whole number of differential genes; *Y*-axis: the name of the enriched KEGG pathways. The sizes of the dots represent the count of the differential genes, and the color of the dots represents the significant enrichment from red to blue.

**Figure 6 animals-13-02254-f006:**
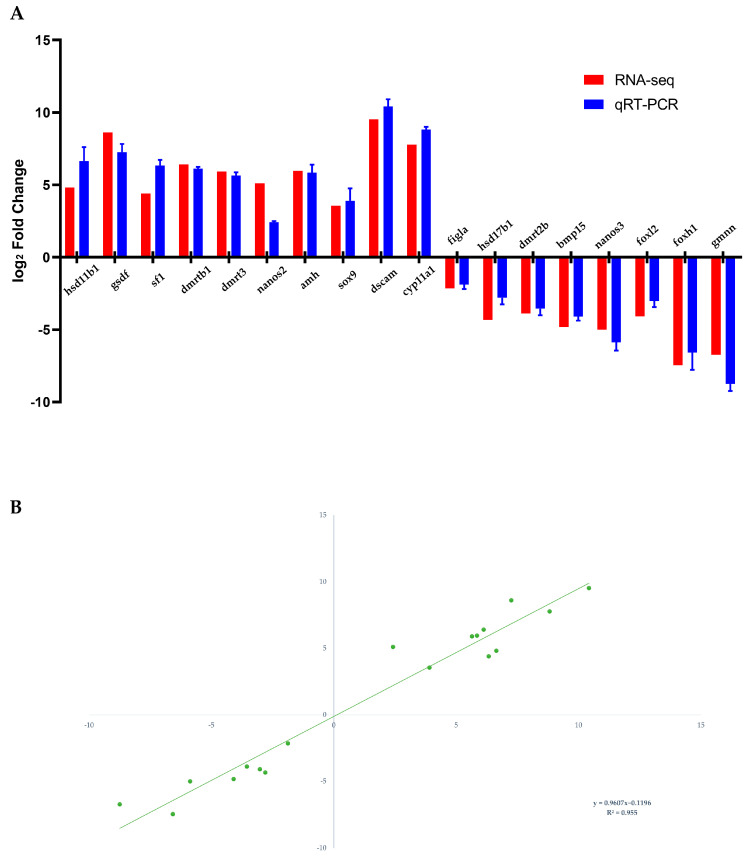
Expression patterns verification of 10 testes-biased and eight ovaries-biased genes (**A**) by qRT-PCR. Correlation analysis of the RNA-Seq data and qRT-PCR data (**B**).

**Figure 7 animals-13-02254-f007:**
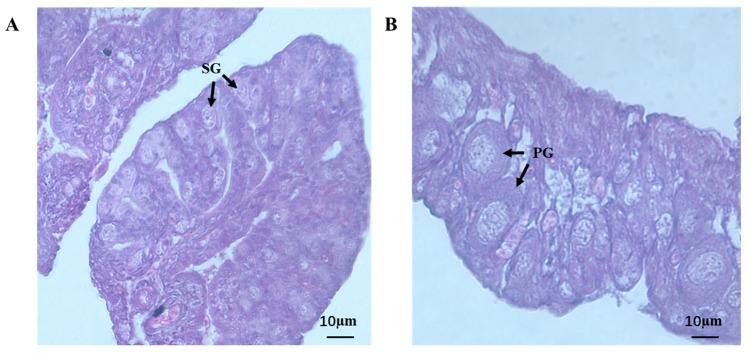
Testis (**A**) and ovary (**B**) histology of jade perch. SG: primary spermatogonia; PG: primary growth oocyte. Scale bars: 10 μm.

**Table 1 animals-13-02254-t001:** qRT-PCR primers of jade perch.

Gene	Sequence (5′-3′)		Product Size (bp)
	Forward Primer	Reverse Primer	
*hsd11b1*	TCCTCGTGTACTCATCTTG	ACCTAATGGCTATTGGTG	167
*gsdf*	AGTAATGCCCGTGTTGTG	GCATCCTGGACATTGGTG	157
*sf1*	CTGTTCAAATGTTGGGAGAC	AGGTGAGCAGACCGTAGTG	135
*dmrtb1*	CCTGTGATTCACTTTCCGTTTA	GAGGTGCGGGTTCTGGTT	196
*dmrt3*	AGCCCAAATCTTCATTTCATG	GGGACACTTTCGGAGGTCA	146
*nanos2*	AACAGCCTCCATCGTGAA	TGCCCTGGAATAAAGTGTC	124
*amh*	TAAGTCCCGTGCTCATTC	AACACCGCCAACATCTAC	182
*sox9*	AGGAGGCTGAGCGTTTGA	CCTTGAAGATCGCATTTGG	147
*dscam*	GGGTTCTTCGGGCTTACA	TTTGCCGCTGGTCCTATT	193
*cyp11a1*	AATAGCCGAGACCGAGAT	TGGTGTTACCGATTAGTATG	165
*figla*	TCGCCGAGGACTTCAATG	ATTCCGTCGCAGCCTTTA	171
*hsd17b1*	TACTCATCCCTCCGTTGC	AGTCACTCCCTCCTGTTG	200
*dmrt2b*	TACTCACGCCAACTCTGTC	CACTATGCTGCTAACCATTT	125
*bmp15*	TCTCGAAGTCCCGTCTGTT	AAGGGTCTTTGGGCTCTG	173
*nanos3*	CCTCCAGACTCGTTTCGC	GACTCACCGTTGTGCTTG	191
*foxl2*	GCCTCGGTGTTGTAGTCAT	GCAACGGTCAGGATAAGC	162
*foxh1*	TTATCAGACCAGCCTTTG	GTGTCCGTGTTTCAGTTT	132
*gmnn*	AAACAGTGGAGCACAGAA	GTAGGTGGAAGGAGGAGT	150
*β-actin*	TGCTGTCCCTGTATGCCTCTGG	TGATGTCACGCACGATTTCCCT	230

**Table 2 animals-13-02254-t002:** Summary statistics of the gonadal RNA-seq data.

Sample	Reads Number	Total Base	GC Content (%)	Q20 (%)	Q30 (%)
Ovary 1	45,819,196	6,872,879,400	51.05	97.52	93.09
Ovary 2	52,776,122	7,916,418,300	51.30	97.46	92.96
Ovary 3	44,164,748	6,624,712,200	51.27	97.38	92.65
Testis 1	38,521,802	5,778,270,300	48.68	97.77	93.51
Testis 2	43,325,712	6,498,856,800	48.64	97.57	93.01
Testis 3	49,925,956	7,488,893,400	49.13	98.57	95.53
Mean	45,755,589	6,863,338,400	50.01	97.71	93.46
Total	274,533,536	41,180,030,400			

**Table 3 animals-13-02254-t003:** Summary statistics of the jade perch gonadal transcriptome assembly and annotation.

Database	Number
Assembly	
Gene Number (#)	107,060
Total length (nt)	107,070,530
Average length (nt)	1000
Max length (nt)	18,090
Min length (nt)	185
N50 (nt)	2336
GC	45.09%
Annotation	
Total number of annotated unigenes	28,514
Unigenes match against NR	27,893
Unigenes match against UniProt	28,165
Unigenes match against KEGG	17,300
Unigenes match against KOG	15,592
Unigenes match against GO	21,465

**Table 4 animals-13-02254-t004:** Summary statistics of DEGs in male (testes) and female (ovaries) jade perch.

Testes VS Ovaries	Number of DEGs
UP	14,332
Down	9517
Total	23,849

**Table 5 animals-13-02254-t005:** Searching for DEGs putatively involved in reproduction from testes and ovaries transcriptome of jade perch.

Unigene ID	log2 Fold Change (Testes vs. Ovaries)	*p*-Value (Testes vs. Ovaries)	FDR (Testes vs. Ovaries)	Gene Annotation	Gene Name
unigene016557	−3.401	6.81 × 10^−46^	3.80 × 10^−44^	Cytochrome P450 family 26 subfamily A	*cyp26a*
unigene078672	−2.938	2.56 ×10^−21^	1.46 × 10^−20^	Cytochrome P450 family 20 subfamily A	*cyp20a*
unigene076774	−2.274	2.41 × 10^−18^	1.08 × 10^−17^	Cytochrome P450 family 27 subfamily C1	*cyp27c1*
unigene017250	−2.206	1.65 × 10^−20^	8.81 × 10^−20^	Cytochrome P450 family 4 subfamily B1	*cyp4b1*
unigene019234	−3.958	1.75 × 10^−29^	2.08 × 10^−28^	Cytochrome P450 Family 19 Subfamily A Member 1	*cyp19a1a*
unigene054857	−3.604	6.40 × 10^−29^	7.21 × 10^−28^	17beta-estradiol 17-dehydrogenase	*hsd17b2*
unigene003247	−4.798	1.83 × 10^−25^	1.50 × 10^−24^	Transcription factor SOX2	*sox2*
unigene009779	−3.106	2.77 × 10^−15^	9.75 × 10^−15^	Transcription factor SOX6	*sox6*
unigene065843	−7.374	6.46 × 10^−22^	3.88 × 10^−21^	Forkhead box protein I1	*foxi*
unigene093705	−4.301	7.81 × 10^−62^	1.79 × 10^−59^	Forkhead box protein H1	*foxh*
unigene044978	−3.426	1.88 × 10^−29^	2.23 × 10^−28^	Forkhead box protein K2	*foxk*
unigene087937	−4.816	4.54 × 10^−30^	5.69 × 10^−29^	Bone morphogenetic protein 15	*bmp15*
unigene074406	−2.156	2.49 × 10^−11^	6.39 × 10^−11^	Bone morphogenetic protein 8	*bmp8*
unigene028905	−4.548	3.01 × 10^−17^	1.23 × 10^−16^	Growth differentiation factor 3	*gdf3*
unigene061344	−6.234	3.08 × 10^−20^	1.61 × 10^−19^	Growth differentiation factor 9	*gdf9*
unigene080631	−2.147	3.33 × 10^−14^	1.08 × 10^−13^	Factor in the germline alpha	*figla*
unigene034002	−6.663	2.89 × 10^−21^	1.64 × 10^−20^	Follistatin-related protein 4	*fstl4*
unigene075890	−2.043	9.42 × 10^−16^	3.43 × 10^−15^	7alpha-diol 3beta-dehydrogenase	*hsd3b7*
unigene033729	−3.509	1.16 × 10^−07^	2.18 × 10^−07^	Insulin-like growth factor 2 mRNA-binding protein 3	*igf2bp3*
unigene052970	−3.108	1.31 × 10^−13^	4.04 × 10^−13^	Insulin-like growth factor 2 mRNA-binding protein 1	*igf2bp1*
unigene103953	−6.835	3.35 × 10^−05^	5.05 × 10^−05^	Zona pellucida sperm-binding protein 3	*zp3*
unigene085743	−2.694	1.55 × 10^−16^	5.98 × 10^−16^	Zona pellucida sperm-binding protein 1	*zp1*
unigene053477	−6.867	5.19 × 10^−04^	7.11 × 10^−04^	Zona pellucida protein C	*zpcx*
unigene033757	−4.007	1.53 × 10^−59^	3.02 × 10^−57^	Cholesterol 25-hydroxylase	*ch25h*
unigene064052	−5.032	2.11 × 10^−34^	3.99 × 10^−33^	Sperm-associated antigen 1	*spag1*
unigene078686	−4.308	5.92 × 10^−19^	2.79 × 10^−18^	Nuclear autoantigenic sperm protein	*nasp*
unigene078850	−4.126	9.62 × 10^−35^	1.87 × 10^−33^	Sperm-associated antigen 5	*spag5*
unigene019789	−2.85	1.31 × 10^−22^	8.31 × 10^−22^	Sperm-sociated antigen 7	*spag7*
unigene006952	−5.014	3.16 × 10^−52^	3.29 × 10^−50^	Paired box protein 7	*pax7*
unigene076579	−2.84	5.79 × 10^−13^	1.70 × 10^−12^	Paired box protein 3	*pax3*
unigene088720	−3.888	7.02 × 10^−19^	3.28 × 10^−18^	Doublesex-and mab-3-related transcription factor 2	*dmrt2*
unigene096363	−4.326	1.36 × 10^−18^	6.20 × 10^−18^	17beta-estradiol 17-dehydrogenase 1	*hsd17b1*
unigene054857	−3.604	6.40 × 10^−29^	7.21 × 10^−28^	17beta-estradiol 17-dehydrogenase 2	*hsd17b2*
unigene076158	−6.687	4.18 × 10^−25^	3.32 × 10^−24^	WNT1-inducible-signaling pathway protein 2	*ccn5*
unigene077407	−4.994	1.62 × 10^−17^	6.78 × 10^−17^	Protein nanos 3	*nanos3*
unigene067731	7.352	4.42 × 10^−08^	8.63 × 10^−08^	Cytochrome P450 family 2 subfamily J	*cyp2j*
unigene077896	3.707	3.22 × 10^−36^	7.09 × 10^−35^	Cytochrome P450 family 1 subfamily C	*cyp1c*
unigene082681	3.625	2.87 × 10^−34^	5.35 × 10^−33^	Cytochrome P450 family 3 subfamily A	*cyp3a*
unigene053000	3.344	1.05 × 10^−07^	1.98 × 10^−07^	Cytochrome P450 family 1 subfamily B1	*cyp1b1*
unigene005081	3.281	4.53 × 10^−07^	8.05 × 10^−07^	Cytochrome P450 family 2 subfamily K	*cyp2k*
unigene045134	2.624	1.31 × 10^−15^	4.72 × 10^−15^	Long-chain fatty acid omega-monooxygenase	*cyp2u1*
unigene042180	2.227	7.98 × 10^−17^	3.14 × 10^−16^	25/26-hydroxycholesterol 7alpha-hydroxylase	*cyp7b*
unigene068680	2.218	2.13 × 10^−04^	3.01 × 10^−04^	Cytochrome P450 family 1 subfamily A	*cyp1a*
unigene073842	10.903	2.66 × 10^−26^	2.36 × 10^−25^	Steroid 21-monooxygenase	*cyp21a*
unigene067853	5.26	4.54 × 10^−41^	1.58 × 10^−39^	3beta-hydroxy-Delta5-steroid dehydrogenase	*hsd3b*
unigene079680	4.966	1.40 × 10^−40^	4.65 × 10^−39^	Steroidogenic acute regulatory protein	*star*
unigene090545	4.923	7.45 × 10^−12^	2.00 × 10^−11^	Cholestenol Delta-isomerase	*ebp*
unigene050885	4.818	4.95 × 10^−34^	9.00 × 10^−33^	Corticosteroid 11-beta-dehydrogenase isozyme 1	*hsd11b1*
unigene052718	4.402	6.14 × 10^−49^	4.49 × 10^−47^	Nuclear receptor subfamily 5 group A member 2	*nr5a2*
unigene009338	3.749	6.39 × 10^−07^	1.12 × 10^−06^	Steroidogenic acute regulatory protein	*star*
unigene091255	2.622	1.22 × 10^−05^	1.91 × 10^−05^	3-oxo-5-alpha-steroid 4-dehydrogenase 2	*srd5a2*
unigene080869	3.559	6.58 × 10^−18^	2.84 × 10^−17^	Transcription factor SOX9B	*sox9b*
unigene027228	3.003	5.24 × 10^−06^	8.48 × 10^−06^	Transcription factor SOX11	*sox11a*
unigene026090	2.897	2.14 × 10^−10^	5.10 × 10^−10^	Transcription factor SOX8	*sox8*
unigene077161	2.82	7.37 × 10^−04^	9.98 × 10^−04^	Transcription factor SOX4B	*sox4b*
unigene009778	2.756	1.90 × 10^−09^	4.19 × 10^−09^	Transcription factor SOX13	*sox13*
unigene027229	2.274	2.58 × 10^−15^	9.11 × 10^−15^	Transcription factor SOX4A	*sox4a*
unigene083320	7.657	4.72 × 10^−11^	1.18 × 10^−10^	Nuclear receptor subfamily 5 group A member 2	*nr5a2*
unigene044549	5.055	1.62 × 10^−10^	3.89 × 10^−10^	Nuclear receptor subfamily 2 group F member 5	*nr2f5*
unigene086247	5.049	1.08 × 10^−14^	3.64 × 10^−14^	Nuclear receptor subfamily 0 group B member 2	*nr0b2*
unigene008230	4.835	1.70 × 10^−28^	1.85 × 10^−27^	Nuclear receptor subfamily 2 group F member 6	*nr2f6*
unigene037568	4.679	4.81 × 10^−43^	2.04 × 10^−41^	Nuclear receptor subfamily 0 group B member 1	*nr0b1*
unigene033981	4.226	2.96 × 10^−19^	1.43 × 10^−18^	Nuclear receptor-interacting protein 2	*nrip2*
unigene069894	2.619	6.39 × 10^−17^	2.54 × 10^−16^	Nuclear receptor-interacting protein 1	*nrip1*
unigene012830	2.196	5.56 × 10^−03^	7.01 × 10^−03^	Nuclear receptor subfamily 1 group D member 1	*nr1d1*
unigene041601	5.965	1.19 × 10^−48^	8.45 × 10^−47^	Anti-mullerian hormone	*amh*
unigene043369	4.353	5.51 × 10^−06^	8.91 × 10^−06^	Retinoic acid receptor alpha	*rara*
unigene034008	3.416	3.76 × 10^−14^	1.21 × 10^−13^	Cellular retinoic acid-binding protein 1	*crabp1*
unigene015271	3.249	8.39 × 10^−08^	1.60 × 10^−07^	Retinoid X receptor alpha	*rxra*
unigene097808	2.100	9.99 × 10^−05^	1.45 × 10^−04^	Retinoic acid receptor gamma	*rarg*
unigene072197	5.474	2.78 × 10^−65^	8.72 × 10^−63^	Insulin-like growth factor 1	*igf1*
unigene048289	4.157	3.18 × 10^−11^	8.10 × 10^−11^	Insulin-like growth factor-binding protein 6	*igfbp6*
unigene064551	3.796	1.02 × 10^−15^	3.71 × 10^−15^	Insulin-like growth factor 2	*igf2*
unigene076631	3.039	1.09 × 10^−08^	2.26 × 10^−08^	Insulin-like growth factor-binding protein 5	*igfbp5*
unigene024310	2.872	1.16 × 10^−20^	6.26 × 10^−20^	Insulin-like growth factor-binding protein 3	*igfbp3*
unigene048431	2.529	8.27 × 10^−09^	1.73 × 10^−08^	Insulin-like growth factor 1 receptor	*igf1r*
unigene036898	6.412	1.09 × 10^−67^	4.21 × 10^−65^	Doublesex-and mab-3-related transcription factor 6	*dmrt6*
unigene018354	5.909	4.37 × 10^−70^	2.17 × 10^−67^	Doublesex-and mab-3-related transcription factor 3	*dmrt3*
unigene021803	2.791	1.48 × 10^−07^	2.75 × 10^−07^	Star-related lipid transfer protein 10	*stard10*
unigene061776	4.711	4.52 × 10^−46^	2.58 × 10^−44^	spermine oxidase	*smox*
unigene021908	3.402	1.03 × 10^−42^	4.16 × 10^−41^	Follistatin-related protein 3	*fstl3*
unigene025700	2.978	2.51 × 10^−32^	3.87 × 10^−31^	Follistatin-related protein 1	*fstl1*
unigene067853	5.260	4.54 × 10^−41^	1.58 × 10^−39^	3 beta-hydroxy-Delta 5-steroid dehydrogenase	*hsd3b*
unigene040047	9.351	3.11 × 10^−21^	1.76 × 10^−20^	Fibroblast growth factor 1	*fgf1*
unigene013299	7.956	1.96 × 10^−12^	5.54 × 10^−12^	Fibroblast growth factor 17	*fgf17*
unigene066839	5.778	6.17 × 10^−26^	5.30 × 10^−25^	Fibroblast growth factor 20	*fgf20*
unigene055225	4.731	2.60 × 10^−55^	3.48 × 10^−53^	Fibroblast growth factor 2	*fgf2*
unigene026434	3.735	3.73 × 10^−09^	8.02 × 10^−09^	Fibroblast growth factor 13	*fgf13*
unigene085827	3.512	6.43 × 10^−06^	1.03 × 10^−05^	Fibroblast growth factor 10	*fgf10*
unigene032503	3.467	1.09 × 10^−06^	1.87 × 10^−06^	Fibroblast growth factor 24	*fgf24*
unigene091912	3.330	2.74 × 10^−21^	1.56 × 10^−20^	Fibroblast growth factor	*fgf*
unigene044979	3.115	4.57 × 10^−41^	1.58 × 10^−39^	Fibroblast growth factor receptor 2	*fgfr2*
unigene078396	2.139	7.27 × 10^−09^	1.53 × 10^−08^	Fibroblast growth factor 12	*fgf12*
unigene060382	5.109	6.71 × 10^−38^	1.72 × 10^−36^	Protein nanos 2	*nanos2*
unigene050951	4.940	5.29 × 10^−48^	3.56 × 10^−46^	Follicle-stimulating hormone receptor	*fshr*
unigene074397	3.471	1.98 × 10^−13^	6.04 × 10^−13^	Gonadotropin-releasing hormone receptor	*gnrhr*
unigene009338	3.749	6.39 × 10^−07^	1.12 × 10^−06^	Steroidogenic acute regulatory protein	*star*
unigene035083	2.584	4.04 × 10^−19^	1.93 × 10^−18^	Wilms tumor protein 1	*wt1*

## Data Availability

The raw data are available from the SRA (http://www.ncbi.nlm.nih.gov/sra/, accessed on 11 April 2023) data repository (accession number: PRJNA946135).

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
