# Peer review of "De Novo Assembly, Characterization and Comparative Transcriptome Analysis of the Gonads of Jade Perch (Scortum barcoo)"

_animals, 2023, doi:10.3390/ani13142254_

Round 1
Reviewer 1 Report
The authors of De Novo Assembly, Characterization and Comparative Transcriptome Analysis of the Gonads in jade perch (Scortum barcoo) performed a RNASeq analysis comparing male and female jade perch in order to identify genes that are differentially expressed between the sexes. The experiment and subsequent analyses are well done and the manuscript is well articulated. There is not a lot of information on sex determination or methods of identification of the sexes juvenile or adult jade perch, making this a worthy and necessary study, especially since the jade perch is an emerging species for aquaculture. However, there are a few concerns that should be addressed before this reviewer would feel comfortable recommending this manuscript for publication.
The major concern this reviewer has is the lack of gonadal staging presented. Gonad development is a dynamic process and gene expression varies dramatically during the different phases of gonadal development and maturation, including transcriptional responses to gonadotropins and sex steroids. In fact there is evidence that traditionally "male" sex steroids promote advancement in female reproductive development in pre-vitellogenic ovaries in some species, and can massively alter the gonadal transcriptome (e.g. Kortner et al. 2008 DOI: 10.1016/j.cbd.2008.04.001; Lokman et al. 2007 DOI: 10.1530/REP-06-0229; Monson et al. 2017, DOI: 10.1093/biolre/iox124). Jade perch appear to reach market size at 6-months of age, but may not yet be sexually mature (Hu et al., 2018, DOI: 10.19080/OFOAJ.2018.08.555743). Thus it would be helpful to have histological staging of the testis and ovarian follicles of the individuals used for the analyses. Many of the genes that were highlighted in this study were also differentially regulated by androgens in immature female coho salmon in the Monson et al. study.
Minor concerns:
1. Cyp19a1a aromatase is discussed at length in the discussion, however it is not highlighted in the DEG table. One would think that aromatase expression would be dramatically different between ovaries and testes as it encodes the protein responsible for conversion of T to E2 and thus is a major driver of female reproductive development. This perhaps would be explainable with gonad histology, if the gonads were immature and not yet responsive to the gonadotropin signal, females may not be producing significant quantities of E2.
2. Many citations in the text read "Error! reference source not found." This should be fixed.
3. The introduction is rather long and reads much more like a review of sex determination. Consider reducing the length and including citations to reviews. Perhaps refer to Nagahama et al., 2021, DOI: 10.1152/physrev.00044.2019, which reviews much of the information in the introduction or Lubzens et al. 2010 DOI: 10.1016/j.ygcen.2009.05.022 for specifically ovarian development and Bhat et al. 2021 DOI: 10.1111/raq.12563 specifically for for testis development, . These reviews may also be helpful for context of your results.
4. This manuscript will need minor copyediting for English language. There are several (not very serious) English language errors.
Very minor editing will be required for this manuscript. There are only a few lines that need additional explanation.
For example - line 40: "Due to it's high level of omega-3, jade perch is believed to be the omega-3 alternative source" This should specify to what species or food jade perch could be an alternative source of omega-3 for.
Reviewer 2 Report
The paper contains very valuable results, a great deal of work has gone into obtaining them. However, in my opinion, the discussion should be thoroughly revised, as in the current version it is mainly a literature review. Some large parts of the discussion are like chapters in a textbook and not like chapters in a scientific article.
In many places the discussion is a mess, e.g. lines 422 - 432: first there is about sex determination in teleosts, then in male mice, then again in zebrafish, then about asymmetric development of embryos although the chapter title suggests gametogenesis.
line 435: oocytes at phase I and II in the ovary – the authors do not explain which phases they mean? meiosis I and II?
The paper does not clearly state the purpose of the work. Therefore, it is difficult to find in the discussion a connection with the aim of the paper.
Round 2
Reviewer 2 Report
Authors added gonads histology analysis, but this analysis is not correct.
What stage of spermatogenesis do the authors have in mind when they use the term: primitive spermatogonia? Are they spermatogonial stem cells? Or primary spermatogonia (spermatogonia I)?
The cells in the ovary are incorrectly identified and signed; figure 7b does not show vitellogenic oocytes! only primary oocytes, also perinuclear oocytes. Vitellogenic oocytes should contain a yolk, which is usually stained with eosin in reddish-maroon colour!
I encourage you to read and analyse the relevant literature, which will help, e.g.
Brown – Peterson et al. 2011 http://dx.doi.org/10.1080/19425120.2011.555724
Lowerre-Barbieri et al. 2011 DOI: 10.1080/19425120.2011.555725
Lubzens et al. 2010 doi:10.1016/j.ygcen.2009.05.022
Schulz et al. 2010 Spermatogenesis in fish
There is no scale in Figure 7.
Author Response
Dear reviewer:
Thank you for your suggestions and comments. It is our negligence for these errors. Refering to the references you list, we have modified all errors you mentioned in the revised manuscript.
First, the stage of spermatogenesis is primary spermatogonia. Secondly, only primary oocytes were labled. Finally, the scale was added in the figure.
Reference:
Brown-Peterson et al. 2011 doi:10.1080/19425120.2011.555724
Lowerre-Barbieri et al. 2011 doi: 10.1080/19425120.2011.555725
Lubzens et al. 2010 doi:10.1016/j.ygcen.2009.05.022
Schulz et al. 2010 doi:10.1016/j.ygcen.2009.02.013
Once again, thank you very much for your comments and suggestions.
Yours
Sincerely
Jianrong Huang
Round 3
Reviewer 2 Report
I agree with the corrections made by the authors and accept the paper